# UNITE: UNIVERSALLY TRUSTWORTHY GNN VIA SUBGRAPH IDENTIFICATION

## ABSTRACT

Graph Neural Networks (GNNs) have become instrumental in modeling graph-structured data, with applications spanning diverse sectors. Despite their prowess, challenges such as susceptibility to adversarial attacks, inherent biases, and opacity in decision-making processes have emerged. While efforts exist to address individual trustworthiness facets like robustness, interpretability, and fairness, a comprehensive solution remains elusive. This study introduces UNITE (UNIversally Trustworthy GNN via subgraph idEntification), a novel end-to-end framework carefully designed to holistically integrate these dimensions. Unlike traditional approaches, UNITE leverages the intricate relationships between these aspects in graph data, presenting optimization goals grounded in information-theoretic principles. Preliminary experiments on real-world datasets indicate that UNITE outperforms existing methods, achieving a harmonious blend of interpretability, robustness, and fairness. This work not only addresses the pressing challenges in GNNs but also sets a new benchmark for trustworthy graph neural networks, paving the way for their broader adoption in critical domains.

## 1 INTRODUCTION

Graph Neural Networks (GNNs) have emerged as a pivotal methodology for modeling intricate graph-structured data, garnering widespread application across a myriad of domains. Whether in bioinformatics networks Kawahara et al. (2017), trading systems Wang et al. (2021), or social analysis Hamilton et al. (2017), GNNs consistently exhibit exceptional performance in tasks such as node classification, link prediction, and graph classification. The prowess of GNNs in assimilating node features, neighboring information, and local graph structures has not only redefined traditional data analysis paradigms but also catalyzed novel innovations in sectors like finance, healthcare, and the social sciences.

While GNNs have garnered significant success across a spectrum of applications, recent investigations have shed light on several critical shortcomings. These include their susceptibility to adversarial attacks and data noise Chen et al. (2018); Luo et al. (2021), potential to inherit or even exacerbate biases from training data Dai et al. (2018); Jiang et al. (2022b); Spinelli et al. (2021), and an inherent opacity in their decision-making processes Luo et al. (2020); Ying et al. (2019); Feng et al. (2023). Such challenges impede the broader adoption of GNNs in sensitive and critical domains like finance and healthcare. In response, the concept of a *trustworthy GNN* has been introduced, attracting increasing attention from the research community. In response to the pressing demand for trustworthy GNNs, a pile of research has been dedicated to enhance the trustworthiness of GNNs from various perspectives Dai et al. (2022). For robustness, solutions have been proposed to shield GNNs from adversarial attacks, enhancing their resilience Wu et al. (2019). To advance explainability, methods have been introduced to elucidate the decision-making processes of GNNs, providing intuitive insights that bridge the gap between complex algorithms and human understanding Ying et al. (2019). Regarding fairness, techniques have been developed to ensure GNNs operate without manifesting biases, aligning with ethical norms and societal values Dai & Wang (2021).

While significant advancements have been made in the domain of trustworthy GNNs, the majority of research has focused on individual facets of robustness, interpretability, or fairness. Notably, only a handful of studies, such as Miao et al. (2022) and Agarwal et al. (2021), have attempted to address these dimensions in tandem. As illustrated in Table 1, no existing approach has achieved a

Table 1: Comparison of Trustworthy Aspects among Different Approaches.

|  | Interpretability | Robustness | Fairness |
|---|---|---|---|
| Graphair | ✗ | ✗ | ✓ |
| FairDrop | ✗ | ✗ | ✓ |
| NIFTY | ✗ | ✓ | ✓ |
| GSAT | ✓ | ✓ | ✗ |
| UNITE | ✓ | ✓ | ✓ |

comprehensive trustworthy GNN that encompasses all three aspects. In real-world applications, a GNN model that prioritizes only one dimension of trustworthiness often falls short of user expectations Dai et al. (2022). This observation prompts us to ask:

*Is it feasible to develop a universally trustworthy GNN that is interpretable, robust, and fair?*

Addressing the question of a universally trustworthy GNN is fraught with challenges. Primarily, the pathways to achieve robustness, interpretability, and fairness are multifaceted. To navigate this complexity, we delve into the intricate relationship between these facets within graph data and advocate for subgraph identification as a unified approach to enhance GNN trustworthiness. In addition, after establishing the optimization goals grounded in information theory, the practical optimization of these objectives presents its own set of challenges. To this end, we introduce tractable bounds for the optimization goals and detail a specific GNN architecture to realize them. In summary, our work presents UNITE, a comprehensive methodology that seamlessly integrates robustness, interpretability, and fairness, setting a new standard for trustworthy GNNs. Our primary contributions are:

- Unraveling the complex relationship between interpretability, robustness, and fairness in graph data and showcasing how subgraph identification can serve as a unified approach.

- Introducing UNITE, a holistic framework that provides tractable bounds for optimization goals and outlines a dedicated GNN architecture.

- Empirically validating our methodology on three real-world datasets, demonstrating its capability to achieve a balanced and advance performance in interpretability, robustness, and fairness, outperforming several baselines.

## 2 PRELIMINARIES

In this section, we lay the foundational concepts that underpin our study, providing clarity on the notations and methodologies employed. We delve into the structure and attributes of graphs, the operational mechanics of Graph Neural Networks (GNNs), and the significance of Mutual Information in understanding relationships within graph data.

**Graph.** Let us consider a graph $\mathcal{G} = \{A, X, S, Y\}$ consisting of $n$ nodes. The adjacency matrix $A \in \{0, 1\}^{n \times n}$ encodes the relationships between nodes, with $A_{ij} = 1$ indicating an edge between nodes $i$ and $j$. Each node is associated with a $d$-dimensional feature vector, collectively represented by the node feature matrix $X \in \mathbb{R}^{n \times d}$, where $X = [x_1, \cdots, x_n]^T$. The vector $S \in \{0, 1\}^n$ captures binary sensitive attributes of nodes, such as gender or race, while $Y \in \{0, 1\}^n$ denotes the binary ground truth labels. Following previous works, we focus on binary sensitive attributes and binary node classification tasks.

**Graph Neural Network (GNN).** GNNs are tailored to process graph-structured data, aptly handling tasks where data entities exhibit intricate interrelationships. A typical GNN model is represented as $f : (A, X) \to \hat{Y}$. The core of GNN operation lies in the aggregation of neighorhood node representations, expressed as: $h_i^{(l+1)} \leftarrow q\left(h_i^{(l)}, \left\{h_j^{(l)} \mid j : A_{ij} = 1\right\}\right)$. Here, $h_i^{(l)}$ denotes the representation of node $i$ at the $l$-th layer, and $q$ is a function that combines information from neighboring nodes.

Through iterative aggregation, GNNs encapsulate both local and overarching graph structures, determining the likelihood of a node's association with a target class.

**Mutual Information (MI).** In the realm of machine learning, Mutual Information serves as a powerful metric to quantify the dependency between two random variables. Given two variables $X$ and $Y$, their Mutual Information $I(X;Y)$ is defined as the difference between the individual entropies and their joint entropy: $I(X;Y) = H(X) + H(Y) - H(X,Y)$. Here, $H(X)$ and $H(Y)$ represent the entropies of $X$ and $Y$ respectively, while $H(X,Y)$ denotes their joint entropy. Intuitively, MI captures the amount of information shared between $X$ and $Y$. In the context of GNNs, MI can be instrumental in understanding the relationship between node features and their corresponding labels, or between nodes and their neighboring structures. A higher MI indicates a stronger dependency, which can be leveraged to enhance the model's predictive capabilities.

## 3 UNIVERSALLY TRUSTWORTHY GNN VIA SUBGRAPH IDENTIFICATION

### 3.1 ON THE COMMONALITIES OF TRUSTWORTHY GNN VIA SUBGRAPH IDENTIFICATION

In this section, we explore the intertwined nature of interpretability, robustness, and fairness in GNNs. We argue that achieving these facets of trustworthiness can be unified under the umbrella of identifying an optimal subgraph within the original graph input.

Interpretability in machine learning seeks to highlight portions of input data that predominantly influence the output. While this has been studied across various data modalities like tabular Sahakyan et al. (2021), image Selvaraju et al. (2017), and text Zhao et al. (2023), for GNNs, it translates to identifying influential subgraphs that significantly contribute to predictions Feng et al. (2023); Ying et al. (2019); Yuan et al. (2021).

Robustness ensures consistent outputs despite noise or adversarial perturbations. Given that real-world graphs often contain noise and irrelevant edges, pruning these edges can enhance GNN robustness Luo et al. (2021); Sun et al. (2022). An optimal subgraph, in this context, is one that retains essential edges while discarding those that introduce noise or vulnerabilities.

Fairness in GNNs is intricately linked to graph topology. Nodes sharing sensitive attributes, like age, often connect more frequently, leading to "topology bias." In GNNs, node representations are aggregated from neighbors, exacerbating this bias Jiang et al. (2022a). By identifying and potentially removing edges that reinforce such biases, we can identify a subgraph that contributes towards a more fair representation.

In light of these observations, it becomes evident that the goals of interpretability, robustness, and fairness in GNNs converge towards the identification of an optimal subgraph. This insight forms the foundation of our approach to achieve a universally trustworthy GNN.

### 3.2 TRUSTWORTHY SUBGRAPH IDENTIFICATION

Various methods have been proposed to identify optimal subgraphs for different aspects of trustworthiness in GNNs. For instance, the combination of Monte Carlo tree search and Shapley value has been employed to measure subgraph importance Yuan et al. (2021). Additionally, adversarial learning has been a popular approach to address robustness and fairness in GNNs Dai et al. (2018); Günnemann (2022). However, these methods often operate in isolation, lacking a unified framework.

In this study, we advocate for the use of mutual information as a constraint to bridge these disparate methods. Mutual information offers a quantifiable measure of the relationship between variables, making it an apt choice to unify the requirements of interpretability, robustness, and fairness. By leveraging mutual information, we aim to consolidate these aspects under a single, cohesive framework, providing a more holistic approach to trustworthy GNNs.

**Interpretability.** Interpretability in GNNs aims to identify a subgraph $G_S$ from the original input graph $G$ that is most influential in label prediction. Consider the solubility of a molecule: the presence of the hydroxy group -OH often indicates solubility in water, making it a positive label-relevant subgraph.

To quantify the importance of a subgraph, we leverage mutual information, a measure capturing the dependency between variables. Specifically, we want to maximize the mutual information between the subgraph $G_S$ and the label $Y$, as formalized by Ying et al. (2019):

$$\max_{G_S} I\left(G_S; Y\right), \text{s.t. } G_S \in \mathbb{G}_{sub}(G), \tag{1}$$

where $\mathbb{G}_{sub}(G)$ represents the set of all subgraphs of $G$. The mutual information $I(G_S; Y)$ can be decomposed as:

$$I\left(G_S; Y\right) = H(Y) - H(Y|G_S) = H(Y) + \mathbb{E}_{Y|G_S}\left(log\mathbb{P}(Y \mid G_S)\right), \tag{2}$$

Here, $H(Y)$ denotes the entropy of label $Y$, a measure of uncertainty. By maximizing $I(G_S; Y)$, we reduce the uncertainty in predicting $Y$, ensuring accurate predictions given the optimal subgraph.

**Interpretability & Robustness.** Robustness in GNNs is defined by the model's resilience against noisy input data. Ideally, the prediction, denoted as $\hat{Y}$, should remain stable despite any task-irrelevant disturbances in the graph, represented as $G_n$. Formally, this is expressed as $G_n \perp\!\!\!\perp \hat{Y}$, signifying the independence between the noise $G_n$ and the prediction $\hat{Y}$.

Given the learning procedure of $G_S$ follows the Markov Chain $< (Y, G_n) \rightarrow G \rightarrow G_S \rightarrow \hat{Y} >$, the relationship $I(\hat{Y}; Y, G_n) \leq I(G_S; Y) \leq I(G_S; Y, G_n)$ is established. This implies that merely maximizing $I(G_S, Y)$ does not ensure robustness. Further, we can derive the following relationship:

$$I\left(G_n; \hat{Y}\right) \leq I\left(G_S; G\right) - I\left(G_S; Y\right). \tag{3}$$

The proof for this inequality is detailed in the Appendix A. This relationship underscores the idea that while maximizing $I\left(G_S; Y\right)$, constraining the information the subgraph $G_S$ inherits from $G$ can reduce the impact of task-irrelevant information on the prediction, thereby bolstering robustness. By integrating Eq. 1 and Eq. 3, we observe that the first two terms form the information bottleneck Tishby et al. (2000). This bottleneck has been previously identified as a key factor in achieving interpretability and robustness in GNNs Luo et al. (2021); Miao et al. (2022).

**Interpretability & Robustness & Fairness.** In the realm of machine learning, fairness aims to mitigate biases, especially concerning sensitive attributes. Here, we emphasize group fairness, which seeks to ensure that individuals within different protected groups receive statistically similar treatments. One common metric for group fairness is statistical (or demographic) parity, which demands that predictions remain independent of the sensitive attribute. This can be quantified as:

$$\Delta SP = \left|\mathbb{P}\left(\hat{Y} = 1 \mid S = 0\right) - \mathbb{P}\left(\hat{Y} = 1 \mid S = 1\right)\right|, \tag{4}$$

where $\hat{Y} = f(G)$ represents the GNN model's prediction given input graph $G$. The goal is to ensure $\hat{Y} \perp\!\!\!\perp S$, indicating prediction independence from the sensitive attribute. However, note that $I(G_S; Y) = I(G_S; Y \mid S) + I(G_S; Y; S)$. This suggests that maximizing $I(G_S, Y)$ might inadvertently introduce the influence of the sensitive attribute $S$ into the subgraph $G_S$. We can relate $\Delta SP$ to mutual information $I\left(G_S; S\right)$ as:

$$I(G_S; S) \geq g\left(\pi, \Delta SP(\mathcal{A}, S)\right), \tag{5}$$

with $\pi = \mathbb{P}(S = 1)$ and $\mathcal{A}$ being any decision algorithm acting on $G_S$. The function $g$ is strictly increasing, non-negative, and convex in $\Delta SP(\mathcal{A}, S)$. A detailed proof is provided in the Appendix B.

Consequently, a subgraph $G_S$ with limited mutual information with $S$ ensures that any decision algorithm based solely on $G_S$ will exhibit bounded parity. The constraint for a fair subgraph becomes:

$$\min_{G_S} I\left(G_S, S\right). \tag{6}$$

In summary, a trustworthy subgraph, when framed in terms of mutual information constraints, should adhere to:

$$\min_{G_S} -I\left(G_S; Y\right) + \alpha I\left(G_S; G\right) + \beta I\left(G_S; S\right), \text{s.t. } G_S \in \mathbb{G}_{sub}(G), \tag{7}$$

where $\alpha$ and $\beta$ are hyperparameters to balance interpretability, robustness, and fairness.

# 4 OPTIMIZATION OBJECTIVE

Graph data, with its inherent non-Euclidean structure, presents unique optimization challenges. Coupled with the computational complexity of mutual information, directly optimizing Eq. 7 becomes a daunting task. To address this, we detail the process of subgraph generation and introduce tractable variational bounds for Eq. 7, offering a more feasible approach to the optimization objective.

## 4.1 SUBGRAPH IDENTIFICATION

To identify the subgraph, we employ a identification neural network, denoted as $\Phi$, which produces $G_S = \{A_S, X_S\}$ from the input graph $G$, i.e., $G_S = \Phi(G)$. Within this neural network, a crucial component is the GNN-based encoder, represented as $\mathrm{Enc}(\cdot)$. This encoder transforms the input graph $G$ into a hidden representation $H \in \mathbb{R}^{n \times r_d}$, effectively capturing the essential features of nodes in $G$. The subsequent sections will delve into the specifics of how the subgraph is generated based on this representation.

**Subgraph Edge Masking.** To generate the adjacency matrix of the subgraph, we employ a process that masks out edges deemed irrelevant for the task at hand. Specifically, for each edge in $G$, we utilize a multi-layer perceptron (MLP) model, denoted as $\mathrm{MLP}_A$, equipped with a sigmoid activation function. This model processes the hidden representations of the vertices associated with the edge to compute a probability for retaining that edge.

Subsequently, a mask is sampled for each edge based on a Bernoulli distribution. The final adjacency matrix for the subgraph, $A_S$, is derived by element-wise multiplication of this mask with the original adjacency matrix. Mathematically, this procedure is represented as:

$$\widetilde{M_{A_{ij}}} = \sigma\left(\mathrm{MLP}_A\left(H(i,:), H(j,:)\right)\right),\; M_{A_{ij}} \sim \mathrm{Bern}\left(\widetilde{M_{A_{ij}}}\right)\; \text{for all } A_{ij} = 1,\; A_S = A \odot M_A, \tag{8}$$

where $H(i,:)$ denotes the $i$-th row of the hidden representation matrix $H$, representing the hidden state of the $i$-th node. The value $\widetilde{M_{A_{ij}}}$ signifies the probability of retaining the edge connecting nodes $i$ and $j$, while $M_{A_{ij}}$ is the corresponding mask sampled from the Bernoulli distribution. The symbol $\odot$ represents the Hadamard (element-wise) product.

**Subgraph Feature Masking.** To generate the feature matrix of the subgraph, we employ a process that masks out features based on their relevance. Specifically, using the hidden representation $H$, a multi-layer perceptron (MLP) model, $\mathrm{MLP}_X$, equipped with a sigmoid activation function, computes a probability matrix, $\widetilde{M_X}$. A feature mask, $M_X$, is then sampled from a Bernoulli distribution and applied to the original feature matrix. The mathematical format of this procedure is:

$$\widetilde{M_X} = \sigma\left(\mathrm{MLP}_X\left(H\right)\right),\; M_X \sim \mathrm{Bern}\left(\widetilde{M_X}\right),\; X_S = X \odot M_X. \tag{9}$$

It's worth noting that the Bernoulli sampling process in subgraph identification is inherently non-differentiable. To facilitate end-to-end training of the subgraph identification, we employ an approximation technique for the Bernoulli sampling. Specifically, we leverage the Gumbel-Softmax reparameterization trick Jang et al. (2016), a widely-adopted method to approximate discrete distributions in a differentiable manner. In essence, the distribution of $G_S$ generated from $G$ via the neural network $\Phi$ can be represented as $G_S \sim \mathbb{P}_\phi(G_S|G)$.

## 4.2 TRACTABLE BOUNDS

Given the objective in Eq. 7, it becomes imperative to derive tractable bounds for the mutual information terms. Specifically, we aim to maximize the mutual information $I(G_S; Y)$ while minimizing $I(G_S; G)$ and $I(G_S; S)$.

**Lower Bound of $I(G_S; Y)$.** To derive a tractable lower bound for $I(G_S; Y)$, we employ a parameterized variational approximation. This approximation allows us to express the mutual information in a more computationally feasible form:

$$I(G_S; Y) \geq \mathbb{E}_{G_S, Y}\left[\log q_\theta\left(Y \mid G_S\right)\right] + H(Y), \tag{10}$$

where $q_\theta(\cdot)$ represents a parameterized variational approximation of the true conditional distribution $\mathbb{P}(Y \mid G_S)$. In this context, we utilize a GNN model for $q_\theta(\cdot)$, which effectively serves as a predictor. This predictor ingests the subgraph $G_S$ as input and outputs the corresponding label $Y$. Given that $H(Y)$ remains constant, our tractable loss can be articulated as:

$$\min_{\theta,\phi} \mathcal{L}_1 = \min_{\theta,\phi} -\mathbb{E}_{G_S,Y} \left[ \log q_\theta \left( Y \mid \mathbb{P}_\phi(G_S \mid G) \right) \right] \tag{11}$$

**Upper Bound of** $I(G_S; S)$**.** To derive a tractable upper bound for $I(G_S; S)$, we employ the Contrastive Log-ratio Upper Bound (CLUB) approach Cheng et al. (2020). This method provides an upper bound as:

$$\begin{aligned} I(G_S; S) \leq \mathrm{I}_{\mathrm{CLUB}}(G_S; S) :=& \mathbb{E}_{p(G_S,S)} \left[ \log q_\omega(S \mid G_S) \right] \\ & - \mathbb{E}_{p(G_S)} \mathbb{E}_{p(S)} \left[ \log q_\omega(S \mid G_S) \right], \end{aligned} \tag{12}$$

where $q_\omega(\cdot)$ represents a parameterized variational approximation of the true conditional distribution $\mathbb{P}(S \mid G_S)$. The parameter $\omega$ governs this approximation. Essentially, $q_\omega(\cdot)$ acts as a predictor, taking the subgraph $G_S$ as input and producing the sensitive attribute $S$ as output. For this approximation, we utilize a two-layer GNN model. The proof for the inequality $I(G_S; S) \leq \mathrm{I}_{\mathrm{CLUB}}(G_S; S)$ is detailed in the Appendix **??**. The resulting tractable upper bound loss is expressed as:

$$\min_{\phi,\omega} \mathcal{L}_2 = \mathbb{E}_{p(G_S,S)} \left[ \log q_\omega(S \mid \mathbb{P}_\phi(G_S \mid G)) \right] - \mathbb{E}_{p(G_S)} \mathbb{E}_{p(S)} \left[ \log q_\omega(S \mid \mathbb{P}_\phi(G_S \mid G)) \right]. \tag{13}$$

**Upper Bound of** $I(G_S; G)$**.** To derive a tractable upper bound for $I(G_S; G)$, we formulate the objective as:

$$\min_{\phi,\tau} \mathcal{L}_3 = \mathbb{E}_{p(G_S,G)} \left[ \log q_\tau(G \mid \mathbb{P}_\phi(G_S \mid G)) \right] - \mathbb{E}_{p(G_S)} \mathbb{E}_{p(G)} \left[ \log q_\tau(G \mid \mathbb{P}_\phi(G_S \mid G)) \right]. \tag{14}$$

The function $q_\tau(G \mid G_S)$ is designed to capture the conditional distribution of $G$ given $G_S$. It is defined as:

$$q_\tau(G \mid G_S) = \sigma \left( \mathrm{MLP}\left( \mathrm{Enc}(G) \oplus \mathrm{Enc}(G_S) \right) \right) \in [0, 1], \tag{15}$$

where $\mathrm{Enc}(\cdot)$ represents the encoder function, and $\oplus$ denotes the concatenation operation. This design ensures that the function captures the joint characteristics of both $G$ and $G_S$.

In summary, the overall tractable loss for our objective is given by:

$$\min_{\phi,\theta,\omega,\tau} \mathcal{L}_1 + \alpha \mathcal{L}_2 + \beta \mathcal{L}_3. \tag{16}$$

## 5 EXPERIMENT

In this section, we conduct experiments to evaluate the performance of UNITE using three real-world datasets: NBA, Pokec-z, and Pokec-n. Comprehensive details about these datasets can be found in the Appendix.

### 5.1 EXPERIMENTAL SETTINGS

**Datasets.**

*Pokec-z and Pokec-n:* Both datasets are subsets of the larger Pokec social network, the leading social platform in Slovakia. These datasets encompass user features such as gender, age, hobbies, interests, education, and working field. For our experiments, the region feature is designated as the sensitive attribute, while the working field serves as the target label for prediction Dai & Wang (2021).

*NBA:* This dataset, an extension of a Kaggle dataset, comprises information on over 400 NBA basketball players from the 2016-2017 season. The dataset includes attributes like nationality, age, salary, and performance statistics. In our study, the player's nationality is considered the sensitive attribute, and the task is to predict whether a player's salary exceeds the median value Dai & Wang (2021).

**Metrics**

*Model Utility:* For evaluating the prediction performance of node classification tasks, we employ the Area Under the ROC Curve (AUROC).

*Interpretability:* We assess interpretability using a fidelity metric inspired by Yuan et al. (2022). Our fidelity metric is defined as:

$$Fidelity = \text{AUC}\left(\hat{Y}_G, Y\right) - \text{AUC}\left(\hat{Y}_{G_S}, Y\right), \tag{17}$$

where $\hat{Y}_G$ represents the model's prediction with the original graph $G$ as input, and $\hat{Y}_{G_S}$ is the prediction using the subgraph $G_S$. Contrary to traditional post-hoc explanation evaluations, we train separate neural networks on $G$ and $G_S$. A lower fidelity score indicates better explainability, and it can be negative if predictions on $G_S$ outperform those on $G$.

*Robustness:* Robustness is evaluated using AUROC under varying levels of data noise. A model demonstrating robustness will maintain a high AUROC despite input perturbations Xu et al. (2021).

*Fairness:* For fairness assessment, we employ the statistical parity metric as detailed in Eq. 4 Beutel et al. (2017).

**Baselines**

We evaluate UNITE against several state-of-the-art methods to provide a comprehensive comparison. The baselines include:

1. **GSAT** Miao et al. (2022): An interpretable and generalizable graph learning approach that employs a stochastic attention mechanism.
2. **GSAT+Adv**: An extension of GSAT that incorporates an adversarial component to mitigate the influence of sensitive attribute information.
3. **NIFTY** Agarwal et al. (2021): A graph contrastive learning method that utilizes fairness-aware graph augmentations to derive fair and stable representations.
4. **Graphair** Ling et al. (2022): A method that learns fair representations by leveraging automated graph data augmentations.
5. **FairDrop** Spinelli et al. (2021): A heuristic approach that drops edges to enhance fairness in graph representation learning.
6. **GCN and GIN**: Standard vanilla GNN models.

## 5.2 EXPERIMENTS RESULTS

In this section we separately compare the interpretability, robustness and fairness for the baselines and UNITE.

### 5.2.1 INTERPRETABILITY & FAIRNESS EVALUATION

We evaluate interpretability and fairness by contrasting UNITE with various baseline methods. For those baselines capable of generating subgraphs, we compute the fidelity score.

**Result Comparison.** Table 2 showcases the AUROC, Fidelity, and $\Delta SP$ metrics of UNITE relative to the baselines. The primary insights are:

- In terms of utility, both the standard GCN and GIN, along with their GSAT counterparts, consistently rank among the top in AUROC scores. For certain datasets, GSAT outperforms the standard GNN, resulting in a negative Fidelity score. This might be attributed to the effective filtering of task-irrelevant information by GSAT.
- GSAT registers the highest Fidelity score, with UNITE trailing closely. The GSAT+Adv variant, however, scores lower in Fidelity compared to GSAT, indicating that adversarial training might adversely affect utility. FairDrop, which relies on a heuristic for edge removal, records the least Fidelity.
- Regarding fairness, Graphair leads in the $\Delta SP$ metric, followed closely by UNITE. It's worth noting that while Graphair and NIFTY excel in fairness, they lack interpretability features. Interestingly, despite its top Fidelity score, GSAT lags significantly in fairness.

Table 2: Comparisons between our method and baselines in terms of AUROC, Fidelity and Fairness. Best results are in **bold** and second best in **bold underline**.

|  |  |  | Vanilla | NIFTY | Graphair | GSAT | GSAT+Adv | FairDrop | UNITE |
|---|---|---|---|---|---|---|---|---|---|
| NBA | GCN | AUROC↑ | **0.783** | 0.759 | 0.763 | **0.778** | 0.760 | 0.755 | 0.771 |
|  |  | Fidelity↑ | - | - | - | **0.005** | 0.023 | 0.028 | **0.012** |
|  |  | $\Delta SP \downarrow$ | 6.35 | 3.59 | **2.89** | 10.37 | 3.27 | 3.21 | **3.18** |
|  | GIN | AUROC↑ | 0.790 | 0.762 | 0.772 | 0.787 | 0.753 | 0.768 | **0.802** |
|  |  | Fidelity↑ | - | - | - | 0.003 | 0.037 | 0.022 | -0.012 |
|  |  | $\Delta SP \downarrow$ | 9.66 | 3.82 | **2.98** | 12.33 | 4.02 | 3.74 | **3.34** |
| Pokec-z | GCN | AUROC↑ | **0.753** | 0.738 | **0.755** | 0.750 | 0.740 | 0.733 | 0.744 |
|  |  | Fidelity↑ | - | - | - | **0.003** | 0.013 | 0.020 | **0.009** |
|  |  | $\Delta SP \downarrow$ | 8.37 | 3.08 | 3.00 | 10.48 | 3.67 | 4.17 | **2.98** |
|  | GIN | AUROC↑ | **0.779** | 0.758 | 0.761 | **0.768** | 0.758 | 0.751 | 0.763 |
|  |  | Fidelity↑ | - | - | - | **0.011** | 0.021 | 0.028 | **0.016** |
|  |  | $\Delta SP \downarrow$ | 9.93 | 3.97 | **3.10** | 11.19 | 4.28 | 5.87 | **3.37** |
| Pokec-n | GCN | AUROC↑ | **0.748** | 0.733 | 0.737 | **0.759** | 0.738 | 0.741 | 0.746 |
|  |  | Fidelity↑ | - | - | - | **-0.011** | 0.010 | 0.009 | **0.002** |
|  |  | $\Delta SP \downarrow$ | 6.28 | 3.04 | **2.71** | 12.48 | 3.86 | 3.18 | 2.95 |
|  | GIN | AUROC↑ | **0.762** | 0.744 | 0.740 | **0.772** | 0.753 | 0.755 | 0.756 |
|  |  | Fidelity↑ | - | - | - | **-0.010** | 0.009 | 0.007 | **0.006** |
|  |  | $\Delta SP \downarrow$ | 7.27 | 5.18 | **2.88** | 13.58 | 4.14 | 3.47 | **3.18** |

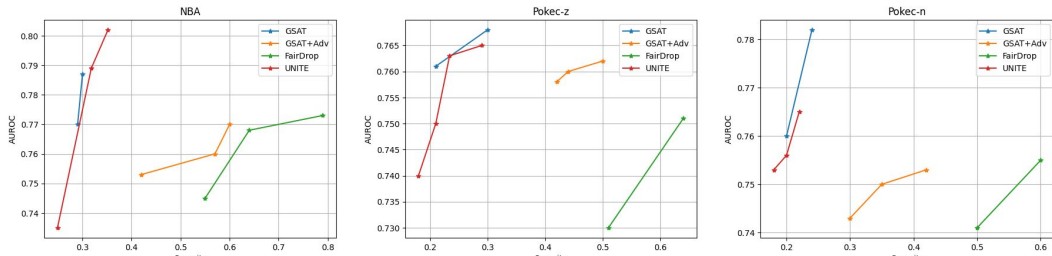

Figure 1: AUROC and Sparsity trade-off on three real-world datasets. Upper-left corner (high AUROC, low Sparsity) is preferable.

**AUROC and Sparsity Comparison.** We also explore the relationship between sparsity and AUROC across the baselines and our method. Sparsity is defined as the ratio of the subgraph $G_S$ size to the original graph $G$ size:

$$Sparsity = \frac{\mid G_S \mid}{\mid G \mid}. \tag{18}$$

An ideal explanatory subgraph $G_S$ should have a low Sparsity score and a high AUROC score. Figure 1 showcases the Pareto front curves generated by hyperparameter grid search for each method. The upper-left corner point represents the optimal performance, with the lowest sparsity and highest AUROC. Results indicate that UNITE offers a competitive AUROC-Sparsity trade-off compared to GSAT, while other fairness-aware baselines (GSAT+Adv and FairDrop) lag behind.

### 5.2.2 ROBUSTNESS EVALUATION

To evaluate robustness, we subject both our method and the baselines to varying levels of edge noise. Specifically, we introduce 20%, 40%, and 60% random edges into the original graph during inference. As observed in Table 3, UNITE may not achieve the highest AUROC score at lower noise rates (0% and 20%). However, at higher noise rates (40% and 60%), UNITE consistently ranks among the top performers, underscoring its robustness. Baselines without explicit robustness considerations (e.g., FairDrop, GCN, GIN, and Graphair) exhibit a decline in AUROC at high noise rates. Although GSAT, a robustness-aware method, demonstrates good robustness, it faces challenges in fairness, as discussed in previous section.

Table 3: Comparisons of our method and baselines under varying input edge noise in terms of AUROC. Best results are in **bold** and second best in **bold underline**.

| | | NBA | | | | Pokec-z | | | | Pokec-n | | | |
|---|---|---|---|---|---|---|---|---|---|---|---|---|---|
| | | 0% | 20% | 40% | 60% | 0% | 20% | 40% | 60% | 0% | 20% | 40% | 60% |
| GCN | Vanilla | **0.783** | **0.773** | 0.732 | 0.688 | **0.753** | **0.748** | 0.731 | 0.711 | **0.748** | 0.740 | 0.726 | 0.703 |
| | NIFTY | 0.759 | 0.750 | 0.742 | 0.733 | 0.738 | 0.737 | 0.730 | 0.721 | 0.733 | 0.729 | 0.719 | 0.704 |
| | Graphair | 0.763 | 0.754 | 0.738 | 0.727 | **0.755** | 0.740 | 0.727 | 0.719 | 0.737 | 0.731 | 0.718 | 0.701 |
| | GSAT | **0.778** | **0.769** | **0.766** | **0.759** | 0.750 | **0.742** | **0.738** | **0.730** | **0.759** | **0.751** | **0.741** | **0.733** |
| | GSAT+Adv | 0.760 | 0.752 | 0.739 | 0.722 | 0.740 | 0.732 | 0.720 | 0.711 | 0.738 | 0.725 | 0.718 | 0.710 |
| | FairDrop | 0.755 | 0.748 | 0.659 | 0.618 | 0.733 | 0.721 | 0.712 | 0.683 | 0.741 | 0.729 | 0.707 | 0.668 |
| | UNITE | 0.771 | 0.769 | **0.763** | **0.751** | 0.744 | 0.740 | **0.735** | **0.729** | 0.746 | **0.741** | **0.733** | **0.712** |
| GIN | Vanilla | **0.790** | 0.768 | 0.721 | 0.618 | **0.779** | **0.771** | 0.751 | 0.730 | **0.762** | **0.757** | 0.732 | 0.711 |
| | NIFTY | 0.762 | 0.760 | 0.755 | 0.721 | 0.758 | 0.755 | 0.747 | 0.738 | 0.744 | 0.739 | 0.728 | 0.715 |
| | Graphair | 0.772 | 0.765 | 0.743 | 0.712 | 0.761 | 0.757 | 0.748 | 0.729 | 0.740 | 0.727 | 0.713 | 0.709 |
| | GSAT | 0.787 | **0.770** | **0.761** | **0.729** | **0.768** | **0.765** | **0.753** | **0.744** | **0.772** | **0.763** | **0.759** | **0.730** |
| | GSAT+Adv | 0.753 | 0.743 | 0.731 | 0.707 | 0.758 | 0.751 | 0.748 | 0.731 | 0.753 | 0.741 | 0.732 | 0.710 |
| | FairDrop | 0.768 | 0.758 | 0.673 | 0.589 | 0.751 | 0.749 | 0.733 | 0.705 | 0.755 | 0.738 | 0.719 | 0.692 |
| | UNITE | **0.802** | **0.787** | **0.779** | **0.731** | 0.763 | 0.760 | **0.757** | **0.749** | 0.756 | 0.750 | **0.744** | **0.739** |

## 6 ABLATION STUDY

In this section, we delve into ablation studies to discern the impact of the robustness constraint, $\mathcal{L}_2$, and the fairness constraint, $\mathcal{L}_3$. We evaluate two variations of UNITE: one without $\mathcal{L}_2$, denoted as "UNITE w/o $\mathcal{L}_2$", and the other without $\mathcal{L}_3$, denoted as "UNITE w/o $\mathcal{L}_3$". We present the results based on the NBA dataset. Table 4 reveals that when omitting $\mathcal{L}_2$, UNITE achieves a commendable $\Delta SP$ but at the expense of utility and robustness. Conversely, excluding $\mathcal{L}_3$ results in impressive utility and robustness but compromises fairness. The full UNITE model strikes a balanced and commendable performance across all three metrics.

Table 4: Ablation study result on NBA dataset. Best results are in **bold** and second best in **bold underline**.

| | AUROC | | | | $\Delta SP \downarrow$ |
|---|---|---|---|---|---|
| | 0% | 20% | 40% | 60% | |
| UNITE | **0.771** | **0.769** | **0.763** | **0.751** | **3.18** |
| UNITE w/o $\mathcal{L}_2$ | 0.764 | 0.753 | 0.741 | 0.737 | **2.05** |
| UNITE w/o $\mathcal{L}_3$ | **0.780** | **0.778** | **0.770** | **0.762** | 9.38 |

## 7 CONCLUSION

In this work, we presented UNITE, a pioneering approach to graph neural networks that harmoniously integrates interpretability, robustness, and fairness. Our methodological framework, grounded in tractable bounds, offers a systematic way to optimize these objectives concurrently. Empirical evaluations on datasets like NBA, Pokec-z, and Pokec-n underscored UNITE's efficacy in striking a balance, a feat often elusive for many existing methods. While our results are promising, future work could delve deeper into enhancing the scalability of UNITE and exploring its applicability in other domains. This research paves the way for more holistic and responsible graph neural network designs in the future.

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

## A  PROOF OF EQ. 3

Given Markov Chain $< (Y, G_n) \to G \to G_S \to \hat{Y} >$, according to data processing inequity, we have:

$$I(G_n; \hat{Y}) \leq I(G_n; G_S), \tag{19}$$

$$I(G; G_S) \geq I(G_S; G_n, Y) = I(G_S; G_n) + I(G_S, Y). \tag{20}$$

Combine Eq. 19 and Eq. 20, we have:

$$I(G_n; \hat{Y}) \leq I(G_n; G_S) \leq I(G; G_S) - I(G_S, Y) \tag{21}$$

## B  PROOF OF EQ. 5

For some $G_S, S \sim \mathbb{P}(G_S, S), G_S \in \mathbb{G}_{sub}, S \in \{0, 1\}$, and any decision algorithm $\mathcal{A}$ that acts on $G_S$, we have

$$I(G_S, S) \geq g\left(\pi, \Delta_{DP}(\mathcal{A}, S)\right)$$

where $\pi = P(S = 1)$ and $g$ is a strictly increasing non-negative convex function in $\Delta_{DP}(\mathcal{A}, S)$.

First, we will show that parity of any algorithm $\mathcal{A}$ that acts on $G_S$ is upper bounded by the variational distance between conditional distributions, $p(G_S \mid S = 1)$ and $p(G_S \mid S = 0)$.

$$\Delta SP(\mathcal{A}, S) = \mid P(\hat{Y} = 1 \mid S = 1) - P(\hat{Y} = 1 \mid S = 0) \mid$$

$$= \left| \int_{G_S} dG_S \, P(\hat{Y} = 1 \mid G_S) \, p(G_S \mid S = 1) - \int_{G_S} dG_S \, P(\hat{Y} = 1 \mid G_S) \, p(G_S \mid S = 0) \right|$$

$$= \left| \int_{G_S} dG_S \, P(\hat{Y} = 1 \mid G_S) \left\{ p(G_S \mid S = 1) - p(G_S \mid S = 0) \right\} \right|$$

$$\leq \int_{G_S} dG_S \, P(\hat{Y} = 1 \mid G_S) \left| p(G_S \mid S = 1) - p(G_S \mid S = 0) \right|$$

$$\leq \int_{G_S} dG_S \left| p(G_S \mid S = 1) - p(G_S \mid S = 0) \right|$$

$$= V(p(G_S \mid S = 0), p(G_S \mid S = 1))$$

where,

$$V(p(G_S \mid S = 0), p(G_S \mid S = 1)) = \int dG_S |p(G_S \mid S = 1) - p(G_S \mid S = 0)| = \|p(G_S \mid S = 1) - p(G_S \mid S = 0)\| \tag{22}$$

is the variational distance between $p(G_S \mid S = 1)$ and $p(G_S \mid S = 0)$. Next, we will show that mutual information, $I(G_S : S)$ is lower bounded by a strictly increasing function of variational distance between $p(G_S \mid S = 1)$ and $p(G_S \mid S = 0)$, and therefore, by transitivity, also lower bounded by the function of parity of any $\mathcal{A}$.

$$I(G_S : S) = \mathbb{E}_{G_S, S} \log \frac{p(G_S, S)}{p(G_S)p(S)}$$

$$= \mathbb{E}_{G_S, S} \log \frac{p(G_S \mid S)}{p(G_S)}$$

$$= (1 - \pi) \mathbb{E}_{G_S \mid S = 0} \log \frac{p(G_S \mid S = 0)}{p(G_S)} + \pi \mathbb{E}_{G_S \mid S = 1} \log \frac{p(G_S \mid S = 1)}{p(G_S)} \tag{23}$$

$$= (1 - \pi) \, \text{KL}(p(G_S \mid S = 0) \| p(G_S)) + \pi \, \text{KL}(p(G_S \mid S = 1) \| p(G_S))$$

$$= \text{JSD}_{(1 - \pi, \pi)}(p(G_S \mid S = 0), p(G_S \mid S = 1))$$

Last step is due to

$$p(G_S) = \sum_S p(G_S, S) = (1 - \pi)p(G_S \mid S = 0) + \pi p(G_S \mid S = 1) \tag{24}$$

and here $\text{JSD}_{(1-\pi,\pi)}(p_1, p_2)$ denotes generalized Jensen-Shannon divergence with mixture weights $(1-\pi, \pi)$. We know that,

$$\text{KL}(p_1 \| p_2) \geq \max\left(\log\left(\frac{2+V}{2-V}\right) - \frac{2V}{2+V}, \frac{V^2}{2} + \frac{V^4}{36} + \frac{V^6}{288}\right) = f(V) \tag{25}$$

For simplicity, we have used $V$ to denote variational distance $V(p_1, p_2)$. $f$ is defined in range $[0, 2)$. We note two important properties of function $f$ that are useful for our proof.

Combining Eq. 23 and Eq. 25, and noting that,

$$V(p(G_S \mid S = 0), p(G_S)) = \|p(G_S \mid S = 0) - p(G_S)\| = \pi\|p(G_S \mid S = 0) - p(G_S \mid S = 1)\|$$
$$V(p(G_S \mid S = 1), p(G_S)) = \|p(G_S \mid S = 1) - p(G_S)\| = (1-\pi)\|p(G_S \mid S = 0) - p(G_S \mid S = 1)\| \tag{26}$$

we get the required result,

$$I(G_S : S) \geq (1-\pi)f(V(p(G_S \mid S = 0), p(G_S))) + \pi f(V(p(G_S \mid S = 1), p(G_S)))$$
$$I(G_S : S) \geq (1-\pi)f(\pi V(p(G_S \mid S = 0), p(G_S \mid S = 1))) + \pi f((1-\pi)V(p(G_S \mid S = 0), p(G_S \mid S = 1)))$$
$$\geq (1-\pi)f(\pi \Delta SP(\mathcal{A}, S)) + \pi f((1-\pi)\Delta SP(\mathcal{A}, S)) \tag{27}$$
$$= g(\pi, \Delta SP(\mathcal{A}, S)) \tag{28}$$

$g$ is a positive weighted combination of non-negative strictly increasing convex functions and therefore also strictly increasing, non-negative, and convex. This completes the proof.

## C    Proof of Eq. 12

To show that $I(G_S; S) \leq I_{CLUB}(G_S; S)$, we calculate the gap between them:

$$\begin{aligned}
\Delta :&= I_{\text{CLUB}}(G_S; S) - I(G_S; S) \\
&= \mathbb{E}_{p(G_S,S)}[\log p(S \mid G_S)] - \mathbb{E}_{p(G_S)}\mathbb{E}_{p(S)}[\log p(S \mid G_S)] \\
&\quad - \mathbb{E}_{p(G_S,S)}[\log p(S \mid G_S) - \log p(S)] \\
&= \mathbb{E}_{p(G_S,S)}[\log p(S)] - \mathbb{E}_{p(G_S)}\mathbb{E}_{p(S)}[\log p(S \mid G_S)] \\
&= \mathbb{E}_{p(S)}\left[\log p(S) - \mathbb{E}_{p(G_S)}[\log p(S \mid G_S)]\right]
\end{aligned} \tag{29}$$

By the definition of the marginal distribution, we have $p(S) = \int p(S \mid G_S)p(G_S)\mathrm{d}G_S = \mathbb{E}_{p(G_S)}[p(S \mid G_S)]$. Note that $\log(\cdot)$ is a concave function, by Jensen's Inequality, we have $\log p(S) = \log\left(\mathbb{E}_{p(G_S)}[p(S \mid G_S)]\right) \geq \mathbb{E}_{p(G_S)}[\log p(S \mid G_S)]$. Applying this inequality to equation (11), we conclude that the gap $\Delta$ is always non-negative. Therefore, $I_{\text{CLUB}}(G_S; S)$ is an upper bound of $I(G_S; S)$. The bound is tight when $p(S \mid G_S)$ has the same value for any $G_S$, which means variables $G_S$ and $S$ are independent.

