# OpenReview forum: "UNITE:Universally Trustworthy GNN Via Subgraph Identification"
_ICLR.cc/2024/Conference — Submitted to ICLR 2024_

### Official Review · Reviewer_XTBN · 2023-10-28

**Soundness:** 3 good
**Presentation:** 1 poor
**Contribution:** 2 fair
**Rating:** 3
**Confidence:** 4

**Summary:**

The authors propose a new objective based on mutual information that can address the three trustworthy aspects of GNNs, including robustness, interpretability, and fairness.

**Strengths:**

- Interesting idea that unify all three trustworthiness aspects in GNNs with mutual information.
- Experimental results show the benefit of the method.

**Weaknesses:**

- The clarity, especially the overall writing of the paper should be greatly improved and polished.
- Many analysis details are missing even in the Appendix. See my questions below.
- The experimental results do not strongly support the claim, especially against GSAT+Adv.
- Maybe I missed it, but the authors seem not to provide the code for reproducibility.

First of all, I think the authors propose a great idea in unifying all three trustworthiness aspects of GNNs together. While I think the author conveys the high-level idea of their approach clearly, the current manuscript seems to be missing many details and contains typos, errors, and inconsistencies in details. See my question list below. I feel the manuscript should be greatly polished before being presented at a conference. Unfortunately, I am not able to justify the correctness of both theoretical and empirical results based on this version.

**Questions:**

1.	Please use \citep{} instead of \cite{} whenever your references are not a part of the sentence. I believe this is mentioned in the manual and should be followed for better readability.
2.	In section 3.1, the paragraph pertaining to fairness. The argument is based on homophily assumption, yet heterophilic cases exist in practice. After reading the whole approach, it seems the homophily assumption is not used. Then why such an example is mentioned?
3.	Why the Markov chain $Y,G_n$ -> $G$? There are also cases that $G$ -> $Y$?
4.	In the proof of eq (5), what are the **two important properties** of $f$?
5.	Also, the equation (25) is neither proved nor giving reference. Why it is true?
6.	Why equation (27) is true? What properties of $f$ are used?
7.	If the final implication of equation (5) is that $I(G_S;S)$ bounds SP, then why not just write it out? I feel it is too vague to present the result in the form of equation (5).
8.	The authors mentioned using the Gumbel-Softmax reparameterization trick for making their identification neural network end-to-end learnable. Can the authors at least write out this detail in the Appendix? If it is already in the Appendix, please add a pointer right after the paragraph of equation (9).
9.	The reference to the Appendix right above equation (13) is broken.
10.	In Table 2, the arrow of the fidelity score is in the opposite direction.
11.   In Table 2, the bold and bold with underline is not consistent. For some rows, we have both, for some others, we have only the bold number and for others, we do not even have a bold number. Why this is the case?

---

### Official Review · Reviewer_cwPo · 2023-10-31

**Soundness:** 3 good
**Presentation:** 4 excellent
**Contribution:** 2 fair
**Rating:** 3
**Confidence:** 4

**Summary:**

In this work, the authors emphasize the importance of trustworhty in GNN learning and accordingly present a subgraph-based approach that integrates interpretability, robustness, and fairness into a unified framework. Experimental results on several datasets verify the effectiveness of their approach.

**Strengths:**

- The authors provide a well-structured and organized paper, making it easy for readers to navigate and understand the key concepts and findings.

**Weaknesses:**

- Limited coverage of related works, with a lack of baselines and important studies. For example, there have been works that leverage IB to enhance GNN interpretability and robustness. Difference summary is needed, also the comparison with these algorithms.

*[1] Graph Information Bottleneck for Subgraph Recognition, ICLR, 2021.*

*[2] Graph Structure Learning with Variational Information Bottleneck, AAAI, 2022.*

- The novelty of this work appears relatively limited, as it is considered incremental. Existing works already investigate graph robustness and interpretability from a information perspective. The primary novelty seems to lie in the consideration of fairness (unified framework). However, the integration of robustness, interpretability, and fairness into a unified framework could be achieved through a simple combination of existing approaches. Results (Table 2, Table 3) show that the proposed model is lower than GSAT in terms of interpretability and robustness, and lower than Graphair in terms of fairness. It raises the question of whether designing a model that directly combines the strengths of GSAT and Graphair would outperform the UNITE.

- It is advisable to offer a more comprehensive description of the experimental settings. This should encompass critical elements, including hyperparameters, learning rates, batchsize, machines, etc. Furthermore, the inclusion of parameter sensitivity experiments and the sharing of code and hyperparameters would greatly bolster the study's reproducibility and overall scientific rigor.

- Please cite GCN and GIN (Section 5.1, Baseline).

- Writing typos, e.g., "Appendix ??" (one line above Eq.13).

**Questions:**

- The statement "Comprehensive details about these datasets can be found in the Appendix." -- Where?

- Eq. 5 is not sufficiently clear. It is necessary to provide practical definitions for the variables $g$ and $\mathcal{A}$.

- Eq. 25 lacks a reference, and it is advisable to include one for clarity and credibility.

- Since the results are very similar to the baselines, have significance tests been conducted to assess the statistical significance of the findings?

- In Section 5.2.1, the explanation for why GSAT performs better is explained as its strong capability to filter task-irrelevant information through stochastic attention. Does this mean stochastic attention is more effective in filtering task-irrelevant information (enhancing GNN explainability) than IB theory?

---

### Official Review · Reviewer_4AEk · 2023-11-01

**Soundness:** 2 fair
**Presentation:** 3 good
**Contribution:** 2 fair
**Rating:** 5
**Confidence:** 3

**Summary:**

This paper introduces UNITE, a new graph neural network approach that aims to improve the fairness and robustness of models when dealing with graph data. UNITE does this by constructing subgraphs and optimizing the mutual information between the subgraphs and labels, while reducing the mutual information between the subgraphs and the original graphs as well as sensitive attributes. The method uses the Gumbel-SoftMax reparameterization technique and parametric variational approximation to achieve the goal through loss function optimization. The results on three real-world datasets are present.

**Strengths:**

1.	The approach uniquely leverages subgraph identification as a unified method to simultaneously address interpretability, robustness, and fairness in GNNs.

2.	The authors evaluated their approach on multiple real-world datasets.

3.	The inclusion of visual displays clearly conveys the experimental findings, simplifying the understanding of the proposed method's performance and advantages for readers.

**Weaknesses:**

1.	The paper could benefit from a deeper exploration of how edge removal in the subgraph affects graph reconstruction loss, especially considering the strong homophilic connectivity of nodes. There's a risk that homophilic edge links might be disproportionately removed, potentially compromising the subgraph’s accuracy and the model's truthfulness.

2.	Inl this paper, the experiment reliance on a single performance and fairness metric, particularly with an unbalanced dataset, may not provide a comprehensive assessment of the model. Incorporating additional performance metrics like the F1-score, and more than one fairness metric, would offer a more holistic evaluation.

3.	While the paper addresses the opacity in GNNs’ decision-making processes, it could provide clearer examples or case studies demonstrating how UNITE improves interpretability.

4.	The paper exist some minor errors, such as the unresolved reference on page six ("Appendix ??.")..

**Questions:**

How does the removal of edges in the subgraph, particularly those connected by strong homophilic links, impact the graph reconstruction loss?

---

### Official Review · Reviewer_Qd3u · 2023-11-05

**Soundness:** 2 fair
**Presentation:** 2 fair
**Contribution:** 2 fair
**Rating:** 3
**Confidence:** 4

**Summary:**

This paper aims to develop a universally trustworthy GNN that is interpretable, robust and fair.  The authors first attempt to unify the objectives of interpretability, robustness and fairness, and then propose a framework to achieve these three trustworthy facets via information theory.

**Strengths:**

1. It is a valid research problem to design a universally trustworthy GNN in a principled way;

2. The proposed method demonstrates improvements in these three aspects.

**Weaknesses:**

1. The proposed relationship unifying interpretability, robustness and fairness is superficial and loose, which cannot really lead to a universal treatment of identifying the optimal subgraph. The discussed relationship from what I see is just the format that each facet concerns some spurious/noisy/biased edges, however the connection of these edges are not revealed. For instance, the edges that cause biases may not necessarily be those causing adversarial vulnerability. Due to the lack of in-depth discussion, the solution is also pretty isolated (e.g., one loss term for one facet).

2. The proposed method has limited technical novelty and contribution. Though the authors try to explain from the information theory perspective, the ultimate solution is still a combination of existing treatments for each facet. And there is not related work section to discuss existing works and differentiate them from this work.

3. The current paper presentation does not well justify/explain several key method design and experiment settings, and needs to be extensively improved for publication. See details in Questions.

**Questions:**

1. The paper over-claims robustness as adversarial perturbations in many places, however, neither the design nor the experiment really deal with adversarial attacks, which could be misleading. Can the authors draw a clear boundary when discussing robustness?

2. When dealing with robustness, the authors establish the mutual information relationship based on a Markov chain, which is not justified. If it is an assumption, why is it reasonable? If it is an observation, is there any evidence to support this observation? The authors should make it clear where this Markov chain comes from. Meanwhile, it is not explained why such a chain will lead to the mutual information relationship.

3. When dealing with fairness, usually mutual information concerns two distributions, then what does it mean by $I(G_s; Y; S)$? How is Eq. (5) used (as later content does not really use $g$ again)?

4. What is the evaluation protocol? The method eventually obtains a $G_s$ generator $P_\phi(G_s|G)$. After obtaining this generator, how is it evaluated to report the metrics in experiment (e.g., using it to process every graph and training a GNN etc)?

5. There is NO related work section. Trustworthy GNN is a broad and active research area, and the authors should provide a comprehensive discussion on them and how this work differentiates them.

6. Given that there are multiple existing works in GNN robustness [1] and fairness [2], why is UNITE only compared with the selected ones?

7. The sparsity definition is misleading. It should be as $1- \frac{|G_s|}{G}$, and higher sparsity indicates more compressed interpretation.

[1] Zhang, Xiang, and Marinka Zitnik. "Gnnguard: Defending graph neural networks against adversarial attacks." Advances in neural information processing systems 33 (2020): 9263-9275.

[2] Wang, Nan, et al. "Unbiased graph embedding with biased graph observations." Proceedings of the ACM Web Conference 2022.

---

### Official Review · Reviewer_mTgk · 2023-11-06

**Soundness:** 4 excellent
**Presentation:** 3 good
**Contribution:** 4 excellent
**Rating:** 8
**Confidence:** 5

**Summary:**

Significant strides have been taken in the realm of reliable Graph Neural Networks (GNNs), but the predominant emphasis of research has been directed towards isolated aspects of reliability, such as robustness, interpretability, or fairness. To establish a holistic and all-encompassing approach to trustworthy GNNs, this paper introduces UNITE, an innovative end-to-end framework meticulously crafted to seamlessly unite these three fundamental dimensions. Extensive experimentation conducted on real-world datasets reveals that UNITE surpasses the capabilities of current methodologies, effectively attaining a harmonious fusion of interpretability, resilience, and equity. This breakthrough represents a pivotal step toward realizing the full potential of trustworthy GNNs, providing a comprehensive solution to address the multifaceted challenges in this field.

**Strengths:**

1.The paper's commendable organization and lucid writing style undoubtedly contribute to its accessibility and overall impact. A well-structured and clearly articulated document is crucial in conveying complex concepts, making it easier for readers to grasp the significance of the research.

2. I sincerely highlight the novelty of this paper, which achieves a comprehensive trustworthy GNN that elegantly encompasses all the three aspects (i.e., robustness, interpretability, fairness). This research paves the way for more holistic and responsible graph neural network designs in the future. The novel approach presented in this paper is a remarkable achievement in the field of trustworthy GNNs.

3.The extensive and meticulously executed experiments conducted to validate UNITE's capabilities are a testament to the paper's rigor. The empirical evidence showcasing UNITE's superior performance in interpretability, robustness, and fairness, when compared to various baseline methods, underscores its practical significance and potential to enhance the state-of-the-art. This robust evaluation bolsters the paper's contribution and reinforces its credibility within the scientific community.

4.I appreciate the importance of the problem studied in this paper, i.e., developing a universally trustworthy GNN, which is interpretable, robust, and fairness.

**Weaknesses:**

1.Enhancing the quality of the writing, particularly in the experiment section, is essential for improving the overall clarity and professionalism of this manuscript. It is highly advisable to dedicate significant effort to meticulously proofread and edit the document in order to eliminate any typographical errors and grammatical mistakes.

2.While the manuscript is generally comprehensive, it might be beneficial to consider incorporating additional experiments that explore other basis models, such as GAT (Graph Attention Network). Although the current research may not be severely lacking without these experiments, including them could potentially offer a more comprehensive and robust analysis, thereby strengthening the paper's contribution to the field.

3.The structure and formatting of Tables 3 and 4 would benefit from further refinement. Be sure to pay close attention to the layout, headings, and labels to ensure that these tables are as informative and user-friendly as possible.

**Questions:**

1.In addition to the critical dimensions of interpretability, robustness, and fairness that have been examined in this paper, it would be valuable to consider other facets of trustworthiness in GNNs. Does there exist other dimensions of trust worthy GNNs in addition to the studied Interpretability, Robustness and Fairness in this paper? Does the proposed model could be easily adapted to other dimensions?

2.For the method mentioned in SUBGRAPH IDENTIFICATION section, I would like to know whether the proposed module is heuristically defined or learnable.

3.The detailed statistics of the used datasets are helpful to understand the experimental results. Please consider adding more information about the datasets.

---

### Meta-Review · Area_Chair_Gpoj · 2023-12-06

**Metareview:**

This paper presents a new graph neural network approach named UNITE, which aims to improve the fairness and robustness of models when dealing with graph data. Overall, this paper is clearly written and well organized. Experiments show the benefits of the proposed method over baselines. Reviewers raised many concerns regarding the novelty, technical contributions, experiments, and paper writing. The authors did not provide any responses to the questions and comments from reviewers.

**Justification For Why Not Higher Score:**

Reviewers raised many valid concerns while the authors did not provide any responses.

**Justification For Why Not Lower Score:**

N/A

---

### Decision · Program_Chairs · 2024-01-16

Reject